# Wigner solids of domain wall skyrmions

Kaifeng Yang [1], Katsumi Nagase[2], Yoshiro Hirayama [2,3], Tetsuya D. Mishima[4], Michael B. Santos[4] & Hongwu Liu [1✉]

Detection and characterization of a different type of topological excitations, namely the domain wall (DW) skyrmion, has received increasing attention because the DW is ubiquitous from condensed matter to particle physics and cosmology. Here we present experimental evidence for the DW skyrmion as the ground state stabilized by long-range Coulomb interactions in a quantum Hall ferromagnet. We develop an alternative approach using nonlocal resistance measurements together with a local NMR probe to measure the effect of low current-induced dynamic nuclear polarization and thus to characterize the DW under equilibrium conditions. The dependence of nuclear spin relaxation in the DW on temperature, filling factor, quasiparticle localization, and effective magnetic fields allows us to interpret this ground state and its possible phase transitions in terms of Wigner solids of the DW skyrmion. These results demonstrate the importance of studying the intrinsic properties of quantum states that has been largely overlooked.

[1] State Key Lab of Superhard Materials, College of Physics, Jilin University, 130012 Changchun, People's Republic of China. [2] Department of Physics, Tohoku University, Sendai, Miyagi 980-8578, Japan. [3] Center for Science and Innovation in Spintronics (Core Research Cluster), Tohoku University, Sendai, Miyagi 980-8577, Japan. [4] Homer L. Dodge Department of Physics and Astronomy, University of Oklahoma, 440 West Brooks, Norman, OK 73019-2061, USA. ✉email: hwliu@jlu.edu.cn

The skyrmion, a topological soliton solution for the description of hadrons in 1960s[1], has attracted renewed interest in recent years due to its potential applications in information and communication technology demonstrated in magnetic materials[2,3]. The magnetic skyrmion is classified by a topological charge $Q = \frac{1}{4\pi} \int \mathbf{n} \cdot \left( \partial_x \mathbf{n} \times \partial_y \mathbf{n} \right) \mathrm{d}x \mathrm{d}y$ counting the number of times that a unit vector $\mathbf{n}$ of the local spin wraps around the unit sphere in a spherically symmetric or combed hedgehog configuration. The symmetric hedgehog skyrmion has also been found in an isotopic (fully spin-polarized) quantum Hall ferromagnet (QHF) at filling factor $v = 1$ of a two-dimensional electron gas (2DEG) in GaAs (called QH skyrmions)[4,5]: it is energetically cheaper to constitute this spin texture than to have single spin flips for the GaAs 2DEG with sufficiently low bare Zeeman energy determined by a small effective $g$ factor of $-0.44$. There is an alternative possibility for the realization of skyrmions once low-lying excited states of the domain wall (DW) as a one-dimensional (1D) object have a nontrivial topological charge (referred to as DW skyrmions)[6-9]: the $xy$ ($z$) component of a unit vector of spins gains a $2\pi$ ($\pi$) phase within the length (width) of this excitation along (across) the DW, defining a mapping topologically equivalent to the combed hedgehog sphere. The ubiquitous nature of the DW may hold promise for the creation and application of skyrmions in nonrelativistic and relativistic systems, which is of considerable importance from both theoretical and practical points of view.

More recently, the DW skyrmion has been demonstrated in chiral magnets with a strong Dzyaloshinskii–Moriya interaction induced by the spin–orbit coupling (SOC)[10-12]. Unlike the magnetic skyrmion that can form a lattice as the ground state, the magnetic DW skyrmion is shown to be low-lying spin excitations. In this study, we present data demonstrating the ground state of DW skyrmions in an Ising (easy-axis) QHF (IQHF). In contrast to the magnetic DW skyrmion, the DW skyrmion in the IQHF is expected to carry an electrical charge due to the identity between topological and electrical charge densities and to dominate the electronic properties of the IQHF as similar to the QH skyrmion[6,13]. Therefore, we combine nonlocal resistance measurements with a resistively detected NMR (RDNMR) technique (hereafter called NRDNMR) as a highly sensitive and minimally invasive approach to determine the DW skyrmion in the simplest IQHF at $v = 2$ of an InSb 2DEG[14,15]. The nuclear spin relaxation (NSR) measurement, a generally accepted methodology for determining the low-frequency spin textures in the QH system (e.g., QH skyrmions)[16], shows an increase in magnetic fluctuations in the DW coupling to the nuclei with decreasing temperature and increasing localization. This strongly suggests that the DW skyrmion may condense into a 1D Wigner crystal stabilized by long-range Coulomb repulsions, which displays both continuous and discontinuous phase transitions depending on the way that the effective magnetic field[17] between two approaching Landau levels (LLs) with opposite spins changes.

## Results and discussion

**Nonlocal resistance measurements.** The nonlocal resistance measurement (see "Methods" section and Supplementary Note 1) is known to play a unique role in the study of edge or surface states[18,19]. In this work, we demonstrate that the nonlocal measurement can be used to determine the intrinsic properties of bulk states. The bulk state under study here is the $v = 2$ IQHF that is formed when the tilt angle $\theta$ (Fig. 1a) is tuned to bring the pseudospin-down [$(n,\sigma) = (0,\downarrow)$] (where $n$ and $\sigma$ are the orbital and spin indices, respectively) and pseudospin-up $(1,\uparrow)$ LLs into degeneracy (Fig. 1b). It is shown in Fig. 1c that a resistance spike in longitudinal resistance $R_{\mathrm{SD},12}$ occurs between two separate LL

peaks (one locates at ~11 T and the other above the maximum field of 15 T in our measurements), signaling the presence of the IQHF[20,21]. Note that, for a simple level crossing, the two approaching LL peaks will merge into one[14]. The IQHF was also identified by the RDNMR measurement of the Knight shift at $v = 2$[15] and by the calculation based on the Hartree–Fock theory (see below). Large Zeeman and cyclotron splittings[22] effectively decouple the edge and bulk channels in the InSb 2DEG with relatively low mobility, making it possible to perform the nonlocal measurement. Although the $v = 2$ IQHF is known to originate from dissipative transport through the DW channels[6,21], its nonlocal transport properties have not been described yet. Figure 1c shows that the resistance of this IQHF consists of fine structures (FSs) in the nonlocal measurement, in contrast to that ($R_{\mathrm{LL}}$) of the LL peak outside the IQHF region. The FSs of two nonlocal configurations $R_{23,14}$ and $R_{14,23}$ are found to be different, however, they obey the reciprocity theorem $R_{\mathrm{kl,mn}}^{+B} = R_{\mathrm{mn,kl}}^{-B}$ (Supplementary Fig. 3) derived from the Onsager relation[23]. Furthermore, the response of all FSs to temperature ($T$) is monotonic and in the opposite direction to that of $R_{\mathrm{LL}}$ (Fig. 1d). The $T$ dependence of the FSs is dominated by the variable range hopping (VRH) mechanism (Supplementary Fig. 4) as similar to that of the spike[22], while the decrease in $R_{\mathrm{LL}}$ at high temperatures originates from the phonon-assisted inter-edge-bulk scattering[24]. From the VRH formula, the $B$ dependence of localization length $\xi$ of the $v = 2$ IQHF for different measurement configurations is plotted in Fig. 1e. Note that the edge states corresponding to the two intersecting LLs become part of an array of the (bulk) domains in the $v = 2$ IQHF[15], and in this case, the current is mainly carried by the edge channel related to the lowest LL with negligible scattering to the bulk state. Therefore, the FSs are believed to be determined by the bulk properties of the IQHF. Moreover, these FSs are different from nonlocal resistance fluctuations observed in the LL peak of quantum wires that are induced by the resonant tunneling between edge states through the bulk localized state[25]. Based on the bulk-edge model with an independent edge- and bulk-components (see ref. [18] and Supplementary Notes 1, 2), the four-terminal resistance is only parametrized by the longitudinal resistance of the bulk state and in this sense, all these resistances (e.g., $R_{23,14}$, $R_{14,23}$, and $R_{\mathrm{SD},12}$) are essentially identical. However, the bulk current $I_i^N$ and the corresponding Hall electric-field $\varepsilon_{\mathrm{H}}$ in each segment $i$ depend on the measurement configurations, accounting for the difference between the spike and its nonlocal counterpart. The presence (absence) of the FSs in the $B$-$\xi$ dependence of $R_{23,14}$ ($R_{\mathrm{SD},12}$) with $\varepsilon_{\mathrm{H}} \sim 0.1\,\mathrm{V\,m^{-1}}$ (~14 V m$^{-1}$) (Supplementary Note 2) in Fig. 1e indicates an electric field-dependent hopping transport; all FSs are broadened at large $\varepsilon_{\mathrm{H}}$ and thus merged into a single peak when the graded percolation problem is considered[26]. This conclusion is further supported by the data in Fig. 1f, where the FSs for $R_{23,14}$ become a single peak as $I_{23}$ increases up to 3.16 μA with $\varepsilon_{\mathrm{H}} \sim 10\,\mathrm{V\,m^{-1}}$ and $I^N \sim 10$ nA (where $I^N$ is the sum of $I_i^N$). In this case, the hopping transport is still in equilibrium (Supplementary Fig. 5), and the segments are electrically linear as evidenced by the fact that $R_{\mathrm{LL}}$ obeys the reciprocity theorem (Supplementary Fig. 6). However, the nonlocal resistance in the IQHF region does not satisfy the reciprocity theorem (Supplementary Fig. 6) because the Overhauser shift induced by dynamic nuclear polarization(DNP) breaks the Onsager relation locally. The effect of the DNP on nonlocal resistances was further investigated by the NRDNMR measurement (see "Methods" section).

The NRDNMR signals of the $v = 2$ IQHF for $R_{23,14}^{\pm B}$ and $R_{14,23}^{\pm B}$ obtained at $I^N \sim 10$ nA and the RDNMR signals of the $v = 2$ IQHF for $R_{\mathrm{SD},12}$ at $I^N \sim 500$ nA are shown in Supplementary

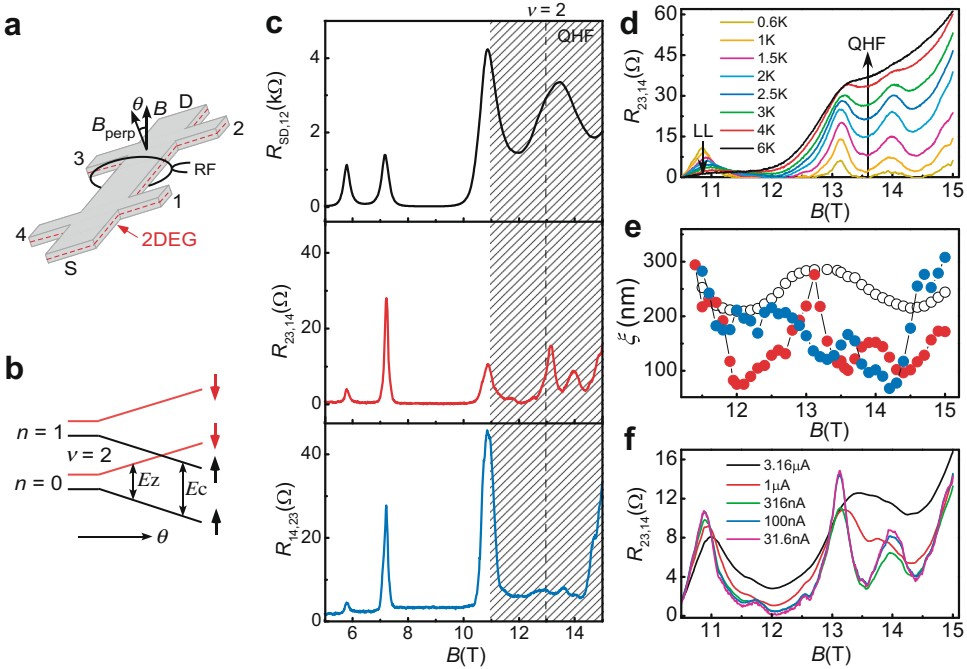

**Fig. 1 Nonlocal resistance measurements of the Ising quantum Hall ferromagnet (IQHF) at filling factor $\nu = 2$. a** Schematics of a two-dimensional electron gas (2DEG, dashed line) Hall bar with six terminals (1–4, S, and D) surrounded by a coil that produces a continuous-wave radiofrequency (RF) field. The tilt angle $\theta$ is defined between the magnetic field $B$ and its perpendicular component $B_{\text{perp}}$. **b** Schematics of the Landau level (LL) splitting as a function of $\theta$. The $\nu = 2$ IQHF occurs when the two LLs with orbital index $n = 0$ and $n = 1$ intersects, where the Zeeman splitting $E_z \sim B_{\text{perp}}/\cos\theta$ and the cyclotron gap $E_c \sim B_{\text{perp}}$ are made equal by adjusting $\theta$. An upward (downward) arrow is for spin-up (spin-down). **c** Four-terminal resistance $R_{kl,mn}$ versus $B$ at $T = 1\,$K, $I_{kl} = 31.6\,$nA, and $\theta = 64°$. The longitudinal resistance $R_{\text{SD},12}$ is given by the ratio of a voltage drop $V_{12}$ between terminals 1 and 2 to current flow $I_{\text{SD}}$ between terminals S and D, and the nonlocal resistance $R_{23,14}$ ($R_{14,23}$) is calculated by $V_{14}/I_{23}$ ($V_{23}/I_{14}$). Note the different scales for longitudinal and nonlocal resistances. The shaded area represents the $\nu = 2$ ($B \sim 13\,$T, dashed line) IQHF region. **d** $R_{23,14}$ versus $B$ at different temperatures. The arrow indicates increasing temperature. **e** Localization length $\xi$ versus $B$ obtained from a fit (Supplementary Fig. 4) to the $B$ dependence of $R_{23,14}$ (red solid dots), $R_{14,23}$ (blue solid dots), and $R_{\text{SD},12}$ (open dots) at different temperatures, respectively. **f** Current ($I_{23}$) dependence of $R_{23,14}$ versus $B$ at $T = 1\,$K.

Figs. 8 and 9, respectively. Note that both RDNMR and NRDNMR signals are expected to be caused by the current-induced DNP[14,27] responsible for the polarization of nuclei in the $\nu = 2/3$ QHF of GaAs 2DEGs, but the underlying mechanism is still unclear[15,16]. A shift of the $\nu = 2$ spike induced by the DNP with the degree of nuclear polarization of ~10% near the DW is assigned to cause the resistance change in the RDNMR measurement of the InSb 2DEG[15]. Relatively low nonlocal resistance (Supplementary Note 2) responds more sensitively to subtle changes in the sample, thus accounting for the DNP performed by very low current in the NRDNMR measurement and for increased detection sensitivity: the maximum signal amplitude for $R_{23,14}^{\pm B}$ and $R_{14,23}^{\pm B}$ is ten times larger than that for $R_{\text{SD},12}^{\pm B}$. Furthermore, the NRDNMR signals for $R_{23,14}^{+B}$ and $R_{14,23}^{-B}$ are found to spread out over the $\nu = 2$ IQHF region but are sparsely distributed for $R_{23,14}^{-B}$ and $R_{14,23}^{+B}$. This suggests that the DW structure in which the DNP occurs depends on the measurement configurations, which is further supported by measurements of nuclear spin-lattice relaxation time $T_1$ and spin dephasing time $T_2$ (see "Methods" section) as local probes of the low-frequency spin dynamics of the DW.

**NRDNMR relaxation time measurements**. We now discuss the $T_1$ and $T_2$ results of the $\nu = 2$ IQHF for $R_{14,23}^{+B}$. Figure 2a shows that $T_1$ near $\nu = 2$ is very short (~8 s) and gradually increases until $|2 - \nu| \sim 0.023$ (dashed line). Then $T_1$ increases sharply on the low (high)-$\nu$ side at 0.3 K (1 K) and decreases with decreasing $\nu$ (data on the high-$\nu$ side are not available). Details of the $T$-

dependent $T_1$ are presented in Fig. 2b in terms of the nuclear spin-lattice relaxation rate $1/T_1$. It is clear that the dependence of $1/T_1$ on $T$ is linear at $|2 - \nu| < 0.023$ (called phase I, Fig. 2a) but following an Arrhenius-like behavior at $|2 - \nu| > 0.023$ (phase II), demonstrating that different relaxation mechanisms are responsible for the two phases. Fast $1/T_1$ at $T \to 0$ in phase I is indicative of strong magnetic fluctuations coupling to the nuclei. Note that $T_1$ in the $\nu = 2$ IQHF obtained from the RDNMR measurement is independent of temperature as also observed in both the $\nu = 2/3$ QHF and the two-subband QHF of the GaAs 2DEG[28]. All these results cannot be explained by a simple level crossing with single spin flips taking account of disorder and electron exchange interaction that accounts for a slow NSR dominated by the Korringa law: $T_1 T = \text{const}$[28]. Furthermore, $R_{14,23}^{+B}$ in phase I is found to be linear in $T$ at large current for the NRDNMR measurement (inset, Fig. 2c), resulting in a linear dependence of $1/T_1$ on $R_{14,23}^{+B}$ (Fig. 2c). Because the localization of quasiparticles in the DW is expected to account for less conducting DWs, an increase in $1/T_1$ with decreasing $R_{14,23}^{+B}$ suggests that quasiparticle localization rather than additional conducting states dominates the NSR in the DW. This is opposite to the Korringa relaxation where $1/T_1$ is proportional to the resistance of the QH state[29]. We point out that these findings are analogous to those of the $\nu \approx 1$ QH state in an extremely high-quality GaAs 2DEG as evidence for the formation of 2D skyrmion crystallization (SC)[30], a Wigner crystal with lattice points occupied by charged skyrmionic spin textures[31]. The DW in the QHF is expected to have electron spins noncollinear to $B$ in the excited states including the spin wave (SW) and the DW skyrmion[6]. The thermally-activated transport

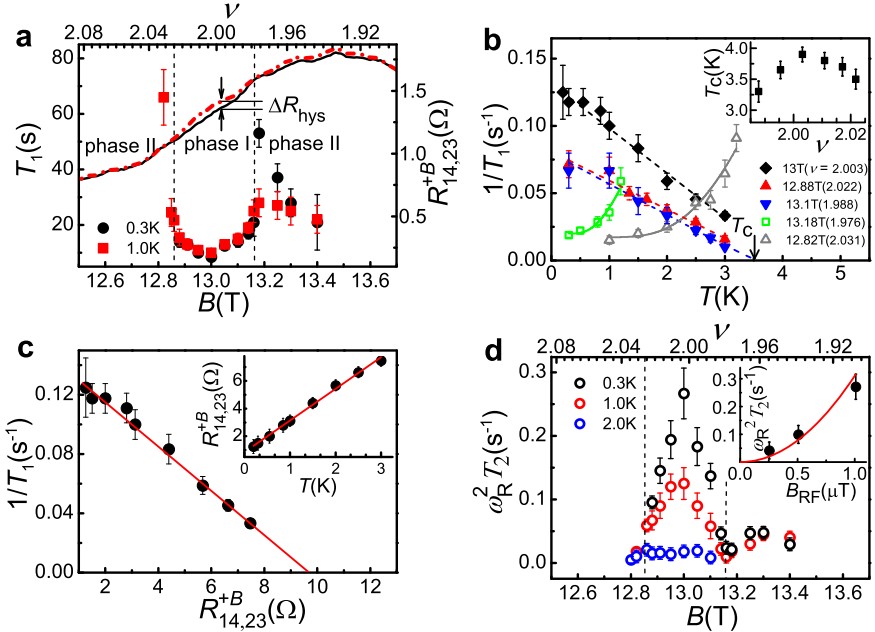

**Fig. 2 Temperature and magnetic field dependence of $T_1$ and $T_2$ in the NRDNMR measurement of the Ising quantum Hall ferromagnet (IQHF) at $\nu = 2$.** **a** $R_{14,23}^{+B}$ (right $y$ axis) as a function of $B$ (or $\nu$) swept upwards (solid line) and downwards (dash-dotted line) with a sweep rate of 1.7 mT s$^{-1}$ at $I_{14} = 3.16 \, \mu A$, $\theta = 64°$, and $T = 1$ K. The $B$ oriented parallel (opposite) to the normal direction of the 2DEG is written as $+(-)B$. $\triangle R_{hys}$ measures the magnitude of the resistance hysteresis at a fixed $B$. The NRDNMR signal occurs around $\nu = 2$ where the $B$ (or $\nu$) dependence of $T_1$ (left $y$ axis) is obtained. The dashed line marks the phase boundary (see main text). Note that error bars for all data in this study are the standard deviation of the mean. **b** $1/T_1$ versus $T$ at different $B$. The solid line is a fit to the data at 12.82 T (13.18 T) using the Arrhenius law with an activation energy $\triangle_{sw}$ of 1.36 (0.42) meV plus a $T$-independent constant of 0.02 s$^{-1}$. The dashed line is a linear fit, and its extrapolation to the $x$ axis defines $T_c$ at which $1/T_1$ is zero. Inset shows the $\nu$ dependence of $T_c$. **c** $1/T_1$ versus $R_{14,23}^{+B}$ obtained by comparing the $T$ dependence of $R_{14,23}^{+B}$ at $B = 13$ T and $I_{14} = 3.16 \, \mu A$ (inset) and that of $1/T_1$ (data at $B = 13$ T, **b**) point by point. The solid line is a guide to the eye. **d** $\omega_R^2 T_2$ versus $B$ (or $\nu$) at different temperatures. The dashed line marks the phase boundary that is the same as that in **a**. The Rabi frequency $\omega_R = \gamma B_{RF}/2$ (where $\gamma = 9.36$ MHz/T is the gyromagnetic ratio of $^{115}$In) is about 10 Hz (see "Methods" section). Inset shows $\omega_R^2 T_2$ as a function of $B_{RF}$ tuned by the RF output power. A good fit ($\propto B_{RF}^2$, solid line) to the data indicates a linear dependence of $\omega_R$ on $B_{RF}$.

measurement of both the $\nu = 2/3$ QHF and the two-subband QHF in the GaAs 2DEG suggests that low-lying excitations akin to the DW skyrmion may favor the NSR, but intricate composite-fermion interactions at $\nu = 2/3$ and the subband degree of freedom complicate the interpretation[28]. Below we show that these excitations are necessarily related to the interpretation and discussion of our data.

The SOC in the IQHF treated as a perturbation due to its off-diagonal entries in the LL basis is expected to twist the planar ($XY$) component of **n** with a 360° rotation along the DW, which lifts the degeneracy of the DW ground state and forms the DW skyrmion excitation[6]. The energy of the DW skyrmion is determined by the SOC strength in the form of $\alpha = \epsilon_{so}/\hbar\omega_c$ [see ref. [6] and parameters given in Supplementary Table 1], in contrast to the case of QH skyrmions whose energy is determined by the effective $g$ factor in the form of the Zeeman energy $E_z$. Note that the QH skyrmion cannot be available in the InSb 2DEG where a relatively large $g$ factor of $-39$ to $-88$[32] makes the ratio of the Zeeman and Coulomb energies 100 times larger than that required for its formation[33]. It is shown in Supplementary Table 1 that $\alpha$ in the InSb 2DEG is only 3.8 times larger than that in the GaAs 2DEG while the difference in $E_z$ between these two 2DEGs is significant (over 100 times). The DW skyrmion in the IQHF is able to carry electric charges similar to the QH skyrmion[13]. The $\nu = 2$ IQHF occurs at zero effective field as discussed later, where dissipative transport in 1D DW channels percolating through the sample is responsible for $R_{14,23}^{+B}$. We expect that single DW skyrmions with long-range Coulomb repulsions will form a crystalline state in the 1D DW channels with quasi-long-range order analogous to the 1D Wigner crystal in carbon

nanotubes[34,35], where a gapless $XY$ spin wave mode with respect to the broken SO(2) symmetry would relax the nuclear spins more efficiently[36,37] as shown by a short $T_1$ at $\nu = 2$ (Fig. 2a). Away from $\nu = 2$ a decrease in the number of DW skyrmions due to a nonzero $b^*$ [21,38] (positive for $\nu > 2$ and negative for $\nu < 2$, Supplementary Fig. 10) makes the crystal harder to form. A crystal of the DW skyrmion with low density is expected to melt at low temperatures because it is easier to break their bonds, as indicated in the inset of Fig. 2b where the state of phase I is found to be thermodynamically most stable up to ~4 K at $\nu = 2$. The NSR in less crystalline states is partially suppressed and therefore $T_1$ becomes longer. When moving further into the region of $|2 - \nu| > 0.023$ (phase II), a sudden change in $T_1$ signals a discontinuous transition to what we interpret as the SW-mediated NSR in neutral DWs with $Q = 0$. The Arrhenius-like behavior of $1/T_1$ in phase II (Fig. 2b) indicates that the NSR occurs in low-energy excited states separated from the ground state by an energy gap $\triangle_{sw}$. It is predicted that the neutral DW is preferably supported by a nonzero $b^*$ [21], where symmetry breaking induced by the SO interaction leads to the opening of a gap in the otherwise gapless spectrum of SW excitations[6,39]. The number of neutral DWs increases with increasing $|b|$, resulting in a reduced $\triangle_{sw}$ by the SW–SW interaction[40] (Supplementary Fig. 11) that accounts for a drop in $T_1$ in phase II (Fig. 2a). Because of the difference in the NSR mode for phase I and phase II, the transformation between them is discontinuous that can be viewed as a first-order phase transition. This is responsible for a jump in $T_1$ between the two phases at low $T$ (Fig. 2a). As $T$ increases, the jump is smeared out (Fig. 2a) due to a decrease in $T_1$ in phase II caused by the Arrhenius behavior of thermal activation. Figure 2d

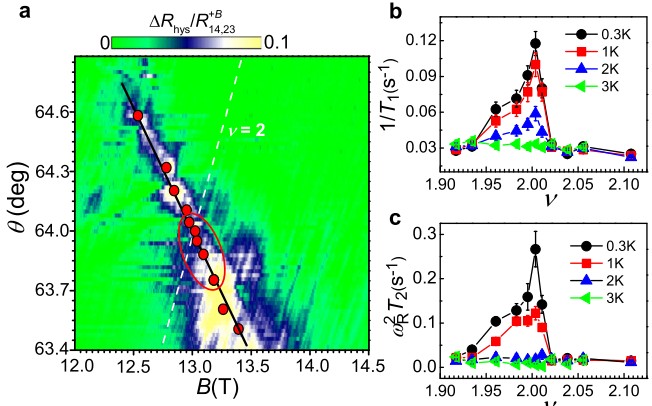

**Fig. 3 $T_1$ and $T_2$ results of the quantum Hall ferromagnet at $\nu = 2$ with zero effective field $b^\star$. a** Contour plot of $B$ and $\theta$ dependence of $\triangle R_{hys}/R_{14,23}^{+B}$ at a field-sweep rate of 1.7 mT s$^{-1}$, $I_{14} = 3.16$ μA, and $T = 1$ K. Red dots indicate the position of maximum $\triangle R_{hys}/R_{14,23}^{+B}$, $1/T_1$, and $\triangle R_{14,23}/R_{14,23}^{sat}$. The solid line is a guide to the eye and the dashed line is calculated for the $B$ and $\theta$ dependence of $\nu = 2$. Temperature dependence of $1/T_1$ (**b**) and $\omega_R^2 T_2$ (**c**) versus $\nu$ where the data (symbols) are in one-to-one correspondence with the red dots in **a** by $\nu = n_s h/eB\cos\theta$ (where $h$ is Planck's constant and $e$ the electron charge). It is clear that the data encircled by an oval in **a** exhibit a strong temperature dependence of $T_1$ and $T_2$.

shows the $T$ and $\nu$ dependence of $T_2$ that is known to be dominated by dynamic fluctuating fields experienced by the nuclei in the quantum well associated with the dynamics of the 2DEG[41]. It is shown that $T_2$ in phase I decreases with increasing $T$ in contrast to the motional narrowing effect, suggesting that the dynamics of the DW skyrmion is in the frozen limit where a very small number of free DW skyrmions suppresses the motion effect[42]. The 1D lattice of localized DW skyrmions may be responsible for the frozen limit that is taken as evidence for the localization of skyrmions in the 2D lattice[41]. In addition, $T_2$ drops off quickly with increasing $|2 - \nu|$ due to relatively strong nuclear dephasing in less crystalline states.

Figure 3a shows a contour plot of $\triangle R_{hys}/R_{14,23}^{+B}$ as a function of $B$ and $\theta$. The position of maximum $\triangle R_{hys}/R_{14,23}^{+B}$ is indicated by solid dots, where both $1/T_1$ and $\triangle R_{14,23}/R_{14,23}^{sat}$ are also at a maximum. The 1D DW channel percolating through the sample is expected to occur at these positions, in which more DW skyrmions are energetically favored. Based on many-body Hartree–Fock theory[17], we have made a calculation of $b^* = 0$ in the $B$-$B_{perp}$ plane (Supplementary Fig. 10) at which the IQHF occurs. It is seen that there is indeed an association between the theoretical $b^* = 0$ line and data points at these positions.

However, the position of these data points is different from that of maximum $R_{14,23}^{+B}$ (Supplementary Fig. 12) that is generally believed to be associated with $b^* = 0$[21]. Our results demonstrate that the resistance measurement cannot establish this association in a straightforward manner. The $T$ dependence of $1/T_1$ and $T_2$ at these positions in terms of $\nu$ is plotted in Fig. 3b, c. Note that these results were obtained along the $b^* = 0$ line rather than away from $b^* = 0$ (Fig. 2). Although the Wigner crystal of the DW skyrmion occurs at $\nu = 2$ with $b^* = 0$, it changes along the $b^* = 0$ line as shown by a nonmonotonic filling dependence of $1/T_1$ and $T_2$ with a peak around $\nu = 2$. This nonmonotonic behavior clearly rules out the possible role of exchange interactions at the Ising transition in $1/T_1$ and $T_2$ because the exchange-correlation energy varies monotonously along the $b^* = 0$ line[20]. Away from $\nu = 2$

backscattering due to a finite density of extended states increases as indicated by an increase in resistance (Supplementary Fig. 13), which effectively induces random potential fluctuations to localize the eigenstates of the DW skyrmion and therefore makes the crystal harder to form. Although the $T$ and $\nu$ dependence of $1/T_1$ and $T_2$ for these less crystalline states are similar to that in Fig. 2, the $\nu$ dependence is highly asymmetric because of the asymmetry of backscattering around $\nu = 2$ as indicated in Supplementary Fig. 13. Deep inside the strong backscattering region the DW skyrmions might have a clumped distribution between the potential wells behaving like a Wigner glass[43], which breaks the quasi-long-range order of the 1D Wigner crystal. The NSR in this region is probably caused by fluctuations of low-lying electronic states with energies smaller than $k_B T$ (where $k_B$ is the Boltzmann constant), exhibiting the temperature-independent $1/T_1$[44]. It is also seen that $T_2$ in the glassy regime is very short and temperature-independent, indicating that more inhomogeneous local field fluctuations prevail over the motion effect for the nuclear dephasing. Note that the $T_1$ and $T_2$ properties of the glassy phase are the same as those of the $\nu = 2$ IQHF for other measurement configurations and also as those of the $\nu = 2$ IQHF obtained from the RDNMR measurement[28], suggesting that the presence of 1D Wigner glass of the DW skyrmion is responsible for the NSR therein. Furthermore, the $T_1$ property specified in the DW Wigner crystal and glass is qualitatively consistent with that predicted in the renormalized classical and quantum critical phases of the skyrmion lattice[45,46], respectively, suggesting that the Wigner solid of the DW skyrmion is able to provide a rich test bed for experimental investigations and theoretical descriptions of quantum criticality.

Note that the width of the DW skyrmion is determined by the DW thickness of the order of magnetic length $l_B = 25.6\text{nm}/\sqrt{B\cos\theta}$ (~11 nm at $\nu = 2$)[15] and its length strongly depends on the SOC strength[13]. The formation of domain structure (several hundreds of nm in domain size[15]) in the InSb 2DEG with relatively low mobility does not allow the DW skyrmion to condense into the Wigner solid in 2D such as the 2D SC observed in a GaAs 2DEG with extremely high mobility[30]. As discussed above, the electrons in the 1D DW channels percolating through the sample move in a correlated fashion, which is expected to dominate the dissipative transport and thus the NSR in the $\nu = 2$ IQHF. It should be pointed out that this 1D skyrmionic Wigner crystal cannot be obtained by laterally confining the 2D SC because of the confinement-induced melting effect[47]. It is also impossible for the skyrmions in 1D nanowires[48] and for the magnetic DW skyrmions[11,12] to condense into a Wigner solid due to the absence of the long-range nature of their interactions. The 1D Wigner crystal consisting of topological solitons instead of electrons is thermally stable up to ~4 K that will be further increased by optimizing the size of the DW skyrmions and thus their interactions, providing a platform where the Wigner crystal can be formed at high temperatures. We note that the 1D skyrmionic Wigner crystal discovered in this study is only distributed over a very narrow range of filling factors ($|2 - \nu|<0.023$) in a certain measurement configuration with nonlocal resistance $R_{23,14}$ but quite robust to condition changes. Crucial to the success of this discovery is the development of the NRDNMR technique that makes it possible to construct and characterize the DW with different configurations under equilibrium conditions. Our results highlight the uniqueness of the NRDNMR technique in the determination of intrinsic properties of quantum states and of their phase transition. Results of the present study strongly suggest that attention should be paid to previous transport and NMR measurements, where the role played by the Hall electric field has tended to be overlooked.

## Methods

**Nonlocal resistance measurement.** The sample of 2DEGs in a 20-nm-wide InSb quantum well under study here was patterned into a Hall bar (100 μm length and 30 μm width) with Ti/Au as Ohmic contacts. A low-noise preamplifier (Stanford Research Systems, Model SR560) and an alternating current (AC) resistance bridge (Lake Shore Model 370) at 13.7 Hz were used for the direct current (DC) and AC measurements, respectively. It is shown in Supplementary Fig. 2 that the AC nonlocal resistance $R_{23,14}$ of the $\nu = 2$ IQHF and its DC counterpart $R_{23,14}^{DC+}$ ($R_{23,14}^{DC-}$) for positive (negative) current satisfy $R_{23,14} \approx \frac{1}{2}\left(R_{23,14}^{DC+} + R_{23,14}^{DC-}\right)$, suggesting that the FSs are independent of the type of electric current. Because the signal-to-noise ratio of the DC NRDNMR measurement is relatively low, all other data shown in this manuscript were collected using the AC resistance bridge in a dilution refrigerator with in situ rotator stage. Low-temperature electron density of $n_s \sim 2.76 \times 10^{15}\,\text{m}^{-2}$ and mobility of $\mu \sim 19.3\,\text{m}^2\,\text{Vs}^{-1}$ of the InSb 2DEG were obtained from the fast Fourier transform (FFT) analysis of low-field Shubnikov-de-Haas (SdH) oscillations and from the value of $R_{SD,12}$ at $B = 0$, respectively.

**NRDNMR, $T_1$, and $T_2$ measurements.** The details of the NRDNMR measurement are shown in Supplementary Fig. 7: a relatively large current is applied to polarize the nuclei in the $\nu = 2$ IQHF region, as indicated by an exponential increase in the nonlocal resistance (for example, $R_{14,23}$) on a time scale of hundreds of seconds. After $R_{14,23}$ becomes saturated, a continuous-wave RF field at a resonance (Larmor) frequency of $f_{NMR} = \gamma B$ is applied to irradiate the 2DEG (on resonance). The RF field is generated by a single coil with 2 turns (a cross-sectional area of 3 mm × 8 mm and a length of 4 mm) surrounding the sample connected to the RF generator using a 50 Ω coax cable. The change in $R_{14,23}$ representing the depolarization of nuclei reaches a maximum $\Delta R_{nl}$ at complete saturation ($R_{nl}^{sat}$) with the rise time $T_r = T_1/\left(1 + \omega_R^2 T_1 T_2\right)$, where $\omega_R$ is the Rabi frequency. The component of the RF field perpendicular to $B$ ($B_{RF}$) is estimated to be ~2.2 μT for an RF output power of 0 dbm at $\theta = 64°$ that gives $\omega_R \sim 10$ Hz. As the RF frequency is detuned from $f_{NMR}$ (off resonance), $R_{14,23}$ is decreased exponentially due to the repolarization of nuclei. The fall time of repolarization gives $T_1$ and thus $T_2$ is obtained from $T_r$.

## Data availability

The authors declare that data supporting the findings of this study are available within the paper and its Supplementary Information files.

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

## Acknowledgements
We thank G. Yusa, K. Muraki, and N. Kumada for helpful discussions. This work was supported by the National Natural Science Foundation of China (11974132 and 11704144), the Jilin Natural Science Foundation (20180101286JC), the Fundamental Research Funds for the Central Universities, the JST-ERATO, the KAKENHI (15H05867 and 18H01811), and the CSRN (Y.H.) and the GP-Spin (K.N. and Y.H.) in Tohoku University.

## Author contributions
H.L. designed and supervised the research. K.Y. performed the experiments and collected the data. K.N. and Y.H. fabricated InSb/AlInSb Hall bars. T.D.M. and M.B.S. grew InSb heterostructures. H.L. and K.Y. analyzed the data. H.L. wrote the manuscript.

## Competing interests
The authors declare no competing interests.
