## [Peer Review File · Nature Communications]

Reviewers' Comments:

Reviewer #1:

Remarks to the Author:

In the manuscript "Wigner solids of domain wall skyrmions" authors claim observation of skyrmions in the QH ferromagnetic transition at $\nu=2$. Authors study nuclear non-local transport and relaxation time of dynamic nuclear polarization. The main observation is that near 13T relaxation time $1/T_1 \propto (T_c - T)$ while $\nu=0.023$ away from it $1/T_1$ has exponential dependence on temperature. These findings has been interpreted as evidence of domain wall skyrmions which also form a Wigner crystal.

To say mildly it is a bold claim for the data presented and a huge leap of faith. First, non-local transport data needs better understanding, e.g. why non-local resistance increase at high T in the vicinity of the transition. That is counter-intuitive to the increase of bulk conduction near the transition. Second, QHF transition seems to happen at slightly higher fields (peak at 13.5T?) than the region where linear-T dependence of $1/T_1$ is observed. Most importantly, nuclear polarization and relaxation via hyperfine interaction requires presence of low energy spin excitation, that can be spin waves or spin level crossing, not necessarily as exotic objects as skyrmions. Change of spin relaxation mechanism may be simply associated with opening a gap between spin subbands as filling factor deviates from $\nu=2$.

To establish existence of a Wigner solid take much more than observation of VRH (and authors do not show $R(T)$ dependence to demonstrate how well it is described by VRH to reader to trust their coherence length data). There is no signature of periodicity in the data presented which make Wigner crystal claim completely unsubstantiated.

To summarize, I do not think the data presented unambiguously supports the claims made in this manuscript.

Reviewer #2:

Remarks to the Author:

In this manuscript Quantum Hall system near filling fraction of 2 is extensively investigated with combination of transport (both local and non-local) and RF manipulation of the nuclear spins. The authors argue that the work offers evidence for the condensation of the domain wall (DW) skyrmions in a quantum Hall Ferromagnet into a 1D Wigner crystal stabilized by long-range Coulomb interaction. This conclusion is so what speculative, as I will elaborate below. Nevertheless, data clearly shows novel phase transition near a filling fraction of 2 and offer definitely present a novel way to probe true ground state properties.

The measurement techniques employed, quality of the data, and novelty of the experimental approach worth publishing in Nature Communication but not in the present form. I believe that substantial modification of the manuscript are required. I will outline detailed questions and concerns below:

I find that abstract and introduction are very hard to read and/or understand by emphasizing way to much the relevance of Skyrmions in different fields of physics, notably high energy theory/ quantum field theory, but failing to give clear intuitive explanation of Skyrmions - as they are observed in the condensed matter systems. One sentence is enough to emphasize importance of Skyrmion in other fields of physics. While they represent abstract mathematical objects in those fields, they can be directly observed in real materials in the context of condensed matter. So please give a visual sketch of a Skyrmionic excitation, describe what these objects are in a simple language; give application example to attract wide interest. Explain why Skyrmions tend to Crystalize - why in 1D only one expects Wigner crystal. Unless these intuitive explanations are offered at the very beginning of the manuscript it is really difficult to appreciate really extensive and novel data presented.

State in the intro in one sentence what direct evidence of 1D Wigner crystallization you found. Lines 60-70 - Describe more clearly what gives non-local measurements unique power to tackle this problem. Please clearly describe the key aspects of this techniques and give some intuitive

explanation of physics of this.

What is the direct evidence in data that the fine structure is dominated by the variable range hopping - just stating a reference is not very clear.

Fig. 2 - is Raby frequency really 10 Hz or there is a confusion in its definition? This cannot be resonant frequency of In nuclei at the fields of the order of 10 T. Please clarify the experimental conditions.

What is the relevance that the discontinuous phase transitions is observed between phase I - II. What does this imply about the nature of these 2 phases was never discussed except stating the nature of it.

The T_1 data implies that in the phase I - $1/T_1$ is proportional to $1/T$ (Fig.2.b) - the implications of this temperature dependence was never discussed in detail and might hold the key to correctly interpreting the nature of the ground state in the phase I. Instead, the conclusion that there is a 1D Wigner crystallization is based on analogy with other systems. This analogy seems plausible - however, one needs to check if it would produce such temperature dependence. This is much more important than looking into T2 processes, as T1 requires energy conservation and is thus much more limiting on possible nature of the ground states.

Dear Dr. Samuel Bladwell, Editor of Nature Communications,

We are grateful to have been given the invitation to submit the revised version of NCOMMS-20-31637-T “Wigner solids of domain wall skyrmions” to *Nature Communications*. We greatly appreciate the two referees for taking the time and efforts necessary to offer their helpful and constructive comments to improve the manuscript during the COVID-19 pandemic. As suggested, we have substantially revised the manuscript to address each of their concerns and to comply with the editorial requests. The major changes include the following:

- ◆ Rewrite the Abstract and Introduction sections.
- ◆ Add four figures: Fig. 2c, Supplementary Figs. 1, 4, and 13 with corresponding analysis and interpretation.
- ◆ Change Fig. 2c in the original version to Fig. 2d and delete Fig. 2d in the original version.
- ◆ Remove Ref. 4 and keep all other references in the original version with a change in their number.
- ◆ Add Supplementary Note 1.
- ◆ Add five references in Supplementary Information.
- ◆ Move all Extended Data Figures to Supplementary Information.

We hope that you and the referees now deem the revised manuscript worthy of publication in your esteemed journal. Below is a point-by-point response to the referees’ comments.

1. RESPONSES TO REFEREE #1’s COMMENTS:

Thank you very much for your efforts in reviewing our manuscript. Below, we provide a point-by-point response to your comments and questions.

COMMENT 1: *Non-local transport data needs better understanding, e.g. why non-local resistance increase at high T in the vicinity of the transition. That is counter-intuitive to the increase of bulk conduction near the transition; Authors do not show $R(T)$ dependence to demonstrate how well it is described by VRH to reader to trust their coherence length data*

RESPONSE: Thank you for raising these questions. As stated in Supplementary Note 2 of the revised version, the four-terminal resistance described by the bulk-edge model of a QH conductor with independent edge-and bulk-current components is approximated by $R_{kl,mn} \propto I^N R/I_{kl}$. In this sense, the resistance between any two terminals (e.g., $R_{23,14}$, $R_{14,23}$, and $R_{SD,12}$) are essentially identical. Temperature dependence of both $R_{23,14}$ (Fig. 1d) and $R_{SD,12}$ (Fig. 1 in Ref. 15) is found to be dominated by the VRH mechanism. A good fit to the data of Fig. 1d using the VRH formula is shown in Supplementary Fig.4 of the revised version, from which the localization length ξ (Fig. 1e) is obtained.

Note that the VRH conductivity $\sigma_{xx}(T) \propto \exp(-\sqrt{T_0/T})$ is related to the longitudinal (Hall)

resistivity ρ_{xx}^N (ρ_{xy}^N) of the N th LL by $\sigma_{xx}^N = \rho_{xx}^N / [(\rho_{xx}^N)^2 + (\rho_{xy}^N)^2]$ with $\rho_{xy}^N \approx 26 \text{ k}\Omega$, so

$R_{kl,mn} \propto R \propto \rho_{xx}^N \propto \sigma_{xx}^N$ for $\rho_{xy}^N \gg \rho_{xx}^N$ in our case. This means that the bulk conduction of the $\nu = 2$ QHF changes with temperature in the same manner as the resistance. This information is provided in the legend of Supplementary Fig.4

COMMENT 2: *QHF transition seems to happen at slightly higher fields (peak at 13.5T?) than the region where linear- T dependence of $1/T_1$ is observed.*

RESPONSE: Thank you for this question. As shown in Fig. 1c, the $\nu = 2$ QHF occurs in a wide range of B from 11 T to 15 T. The linear- T dependence of $1/T_1$ is observed near $\nu = 2$ with zero effective field b^* whose position is different from that of the resistance spike as shown in Supplementary Fig. 13 of the revised version. Furthermore, we have added the statement ‘‘However, the position of these data points is different from that of maximum $R_{14,23}^{+B}$ (Supplementary Fig. 13) that is generally believed to be associated with $b^* = 0$. Our results demonstrate that the resistance measurement cannot establish this association in a straightforward manner’’ in lines 203-205.

COMMENT 3: *Most importantly, nuclear polarization and relaxation via hyperfine interaction requires presence of low energy spin excitation, that can be spin waves or spin level crossing, not necessarily as exotic objects as skyrmions. Change of spin relaxation mechanism may be simply associated with opening a gap between spin subbands as filling factor deviates from $\nu = 2$.*

RESPONSE: Thank you for this question. It’s right that the hyperfine interaction requires low-energy spin excitations to fulfill the electron-nuclear flip-flop process, which is the key to development of the RDNMR technique. Spin-level crossing is just our initial attempts to perform the RDNMR measurement of InSb 2DEGs with comparable Zeeman and cyclotron energies (Ref. 10). However, there were no RDNMR signals observed at these crossings. Although a single broad peak occurs at these crossings, the Coulomb interaction will cause the peak to appear *before* the single-particle LLs cross where two neighboring LL peaks are merged into a broad peak. Therefore, the energy conservation required for the flip-flop process cannot be satisfied at these crossings. Note that these crossings occur at relatively low B . As the field increases, the increased exchange interaction will lead to a first-order phase transition from a paramagnetic state to a ferromagnetic state where domains with different spin polarizations are energetically degenerate. In such a QHF, domain walls are characterized by electron spins *noncollinear* to B that may support the low-energy spin excitation to favor the flip-flop process. The low-lying excited states of the domain wall include the form of spin waves and of topological defects-skyrmions, which are necessarily related to the interpretation and discussion of our experimental data. We note that the spin wave in a uniform (fully spin-polarized) QHF disperses quadratically from the Zeeman splitting and thus cannot accelerate the nuclear spin relaxation at high B . This information is included in lines 156-162.

COMMENT 4: *To establish existence of a Wigner solid take much more than observation of VRH. There is no signature of periodicity in the data presented which make Wigner crystal claim completely unsubstantiated.*

RESPONSE: Thank you for this question. In fact, evidence for the electron solid reported in semiconductor heterostructures has so far been limited to indirect methods (e.g., NMR, transport, microwave, and thermal measurements) and it is not possible to image the real-space structure of it hosted in the 2DEG that is buried several tens of nm below the surface. In this work, we develop a new technique called “NRDNMR” to characterize the DW in a QHF *under equilibrium conditions* confirmed by the results of VRH, which is not available by conventional methods. Furthermore, extensive and detailed analysis of the dependence of T_1 and T_2 on temperature, filling factor, quasiparticle localization (Fig. 2c, new figure), and effective magnetic fields allows us to interpret the ground state of the DW and its possible phase transitions in terms of Wigner solids of the DW skyrmion. A substantial modification of the Abstract, Introduction, and Results sections in the manuscript has been made for better understanding our data presented.

We greatly appreciate your time and efforts for re-reviewing the manuscript.

2. RESPONSES TO REFEREE #2's COMMENTS:

We're pleased to hear that, in your opinion, we present really extensive and novel data and our measurement techniques employed, quality of the data, and novelty of the experimental approach are worth publishing in Nature Communications. We gratefully appreciate your insightful comments that help improve the manuscript. Below is the detailed response to your comments and suggestions.

COMMENT 1: *I find that abstract and introduction are very hard to read and/or understand by emphasizing way to much the relevance of Skyrmions in different fields of physics, notably high energy theory/ quantum field theory, but failing to give clear intuitive explanation of Skyrmions - as they are observed in the condensed matter systems. One sentence is enough to emphasize importance of Skyrmion in other fields of physics. While they represent abstract mathematical objects in those fields, they can be directly observed in real materials in the context of condensed matter. So please give a visual sketch of a Skyrmionic excitation, describe what these objects are in a simple language; give application example to attract wide interest. Explain why Skyrmions tend to Crystallize - why in 1D only one expects Wigner crystal. Unless these intuitive explanations are offered at the very beginning of the manuscript it is really difficult to appreciate really extensive and novel data presented.*

RESPONSE: Thank you very much for your valuable comments and suggestions. We rewrote the Abstract and Introduction sections to address these concerns (please see main text for more details).

COMMENT 2: *State in the intro in one sentence what direct evidence of 1D Wigner crystallization you found.*

RESPONSE: Thank you for this suggestion. The one-sentence statement added in Introduction is that “an increase in magnetic fluctuations in the DW coupling to the nuclei with decreasing temperature and increasing localization suggests that the DW skyrmions may condense into a 1D Wigner crystal stabilized by long-range Coulomb repulsions, which displays both continuous and discontinuous phase transitions depending on the way that the effective magnetic field between two approaching Landau

levels (LLs) with opposite spins changes.”

COMMENT 3: *Describe more clearly what gives non-local measurements unique power to tackle this problem. Please clearly describe the key aspects of this techniques and give some intuitive explanation of physics of this.*

RESPONSE: Thank you for this suggestion. The key aspects of our technique lie in relatively low nonlocal resistances that respond more sensitively to subtle changes in the sample. This makes it possible to measure the effect of dynamic nuclear polarization (DNP) on the nonlocal resistance at low current corresponding to a small Hall electric field in the bulk, thus allowing for the NRDNMR measurement of the DW structure under equilibrium conditions that is not available by conventional methods. This information is included in lines 47-57 with the addition of Supplementary Note 1 and Supplementary Fig. 1 for better understanding the nonlocal resistance measurement.

COMMENT 4: *What is the direct evidence in data that the fine structure is dominated by the variable range hopping - just stating a reference is not very clear.*

RESPONSE: Thank you for this question. The direct evidence in data is that the temperature dependence of nonlocal resistance of the QHF consisting of fine structures in Fig. 1d is well fit by the VRH formula, as shown in Supplementary Fig.4 of the revised manuscript.

COMMENT 5: *Fig. 2 - is Raby frequency really 10 Hz or there is a confession in its definition? This cannot be resonant frequency of In nuclei at the fields of the order of 10 T. Please clarify the experimental conditions.*

RESPONSE: Thank you for this question. It's correct that the Rabi frequency of $\omega_R = \gamma B_{RF}/2$ (where $\gamma = 9.36$ MHz/T is the gyromagnetic ratio of ^{115}In) is not the resonance (Larmor) frequency of $f_{NMR} = \gamma B$. The continuous-wave RF field is generated by a single coil with 2 turns (a cross-sectional area of $3 \text{ mm} \times 8 \text{ mm}$ and a length of 4 mm) surrounding the sample connected to the RF generator using a 50Ω coax cable. It is estimated that the component of the RF field perpendicular to B (B_{RF}) for an RF output power of 0 dbm at the tilt angle of $\theta = 64^\circ$ is about $2.2 \mu\text{T}$ that gives $\omega_R \sim 10 \text{ Hz}$. These experimental conditions are described in “NRDNMR, T_1 and T_2 measurements” of the Methods section.

COMMENT 6: *What is the relevance that the discontinuous phase transition is observed between phase I-II. What does this imply about the nature of these 2 phases was never discussed except stating the nature of it.*

RESPONSE: Thank you for this question. Because of the difference in the mode of nuclear spin relaxation for phase I (a gapless XY spin wave mode in charged DWs) and phase II (a gapped spin wave excitation in neutral DWs), which cannot be continuously transformed one into another, the transformation between them can be viewed as a first-order phase transition. This accounts for a jump in T_1 between the two phases at low temperatures (Fig. 2a). As the temperature increases, the jump is smeared out (Fig. 2a) due to a drop in T_1 in phase II caused by the Arrhenius behavior of thermal activation. This information is included in lines 185-189.

COMMENT 7: *The T_1 data implies that in the phase I - $1/T_1$ is proportional to T (Fig.2.b) - the implications of this temperature dependence was never discussed in detail and might hold the key to correctly interpreting the nature of the ground state in the phase I. Instead, the conclusion that there is a 1D Wigner crystallization is based on analogy with other systems. This analogy seems plausible - however, one needs to check if it would produce such temperature dependence. This is much more important than looking into T_2 processes, as T_1 requires energy conservation and is thus much more limiting on possible nature of the ground states.*

RESPONSE: Thank you very much for your insightful comments. We agree with your point of view regarding the important role of the T_1 results in understanding the nature of the ground state. Therefore, we present the T dependence of $R_{14,23}^{+B}$ and the $R_{14,23}^{+B}$ dependence of $1/T_1$ in Fig. 2c, and discuss about the implication of this temperature dependence to the nature of the ground state of phase I in lines 147-153.

We greatly appreciate your time and efforts for re-reviewing the manuscript.

Reviewers' Comments:

Reviewer #1:

Remarks to the Author:

I read carefully authors response to my and other referee comments and I still not convinced that spin textures need to be invoked in order to explain electron-nuclear interaction.

I would start my objections with a few qualitative point:

1. theoretically, these experiments are in a regime where skyrmions (at least 2D ones) are energetically unfavorable and only topologically trivial excitations are observed. Skyrmions are controlled by parameter $k=E_z/E_c$ and are favorable for $k<0.02$ (see e.g. Phys. Rev. B 90, 085301 (2014)). In InAs with $g>20$ at $B>10T$ $k>0.7$ or >40 higher. In 1D E_c is further suppressed compared to 2D which further increases k and makes formation of 1D skyrmions even less likely.

2. experimentally, nuclear-electron interaction has been studied in GaAs at 2/3 for >20 years and is conventionally assigned to hyperfine interaction which is allowed due to the crossing of spin-up and down energy levels with no need to introduce a spin texture into domain walls.

Here are more specific comments related to the data presented:

"Although a single broad peak occurs at these crossings, the Coulomb interaction will cause the peak to appear before the single-particle LLs cross where two neighboring LL peaks are merged into a broad peak. Therefore, the energy conservation required for the flip-flop process cannot be satisfied at these crossings."

This statement is rather confusing: either you have a gap between spin-up and down states and $R_{xx}=0$ or you have high enough T and/or disorder to mix the states and than $R_{xx}>0$. Either one or the other! Experimentally you see $R_{xx}>0$ which means the two energy levels are near-degenerate (at least the gap is $<$ disorder broadening) and there is no problem with energy conservation. These materials have much higher disorder than GaAs based on the width of the transition.

"As the field increases, the increased exchange interaction will lead to a first-order phase transition from a paramagnetic state to a ferromagnetic state where domains with different spin polarizations are energetically degenerate."

Is there any EXPERIMENTAL evidence of the QHF formation rather than level crossing (apart from the fact that crossing point does not match single-particle crossing prediction, which it is not expected to)? In GaAs QHF state was identified by analyzing Knight shift in NMR.

"In such a QHF, domain walls are characterized by electron spins noncollinear to B that may support the low-energy spin excitation to favor the flip-flop process."

Is there is any evidence of non-collinear spin arrangement in these domain walls (or any other in QHE experiments)? I do not know of either theoretical nor experimental claims. With such a large g-factor in InSb spin textures will be very energetically unfavorable.

"The low-lying excited states of the domain wall include the form of spin waves and of topological defects-skyrmions, which are necessarily related to the interpretation and discussion of our experimental data."

Hyperfine interaction does not require tilted electron spins since at this T and B equilibrium nuclear polarization is just a few percent and nuclear spins point in all directions.

"We note that the spin wave in a uniform (fully spin-polarized) QHF disperses quadratically from the Zeeman splitting"

Lowest spin excitations are not magnons but magneto-rotons with $E < E_z$, see e.g. 10.1103/PhysRevLett.107.06680.

To summarize, I do not think there is enough evidence to claim indirect observation of spin texture formation in this paper. Simple level crossing + disorder should allow direct hyperfine interaction in this system.

Reviewer #2:

Remarks to the Author:

I find this re-submitted version much improved and easier to read. Also authors have well justified all the conclusions and have softened their main claim (as always one cannot ever be 100% sure and can only offer plausible interpretation of the data).

I find this work important and recommend it to be published in Nature Communications.

RESPONSES TO REVIEWER COMMENTS

1. RESPONSES TO REVIEWER #1's COMMENTS:

Thank you very much for taking the time and effort to carefully read our responses and to let us know your concerns. Below we provide a point-by-point response to these concerns, where the papers of my coauthors and me are marked in red.

COMMENT 1: *Theoretically, these experiments are in a regime where skyrmions (at least 2D ones) are energetically unfavorable and only topologically trivial excitations are observed. Skyrmions are controlled by parameter $k=E_z/E_c$ and are favorable for $k<0.02$ (see e.g. Phys. Rev. B 90, 085301 (2014)). In InSb with $g>20$ at $B>10T$ $k>0.7$ or >40 higher. In 1D E_c is further suppressed compared to 2D which further increases k and makes formation of 1D skyrmions even less likely.*

RESPONSE: We should point out that the skyrmion you mentioned is known as the quantum Hall (QH) skyrmion proposed in 1993 [PRB 47, 16419(1993)]. The most typical example is the QH skyrmion formed in a fully spin-polarized (isotropic) QHF at $\nu = 1$ of a 2DEG in GaAs; the spins in the skyrmionic texture have opposite orientation at the origin and infinity, and have a vortex-like configuration at intermediate distances in a real space. It is energetically cheaper to constitute this spin texture than to have single spin flips for the GaAs 2DEG with sufficiently low bare Zeeman energy E_z due to a relatively small g factor of -0.44, because the Coulomb energy induced by one-electron excitation can be lowered by exciting neighboring electrons coherently that makes the charge spread over a wide domain. Since the excess Zeeman energy of a Skyrmion may compensate the gain in the Coulomb energy due to a large number of flipped spins involved, the 2DEG with a small E_z is requisite for the stabilization of the QH skyrmion. It was predicted [PRB 50, 11018 (1994)] that the QH skyrmion was favorable for $k < 0.022$ as similar to your estimation. The QH skyrmion has been extensively studied in a ferromagnetic state near odd-integer or some fractional filling factors where the electron interaction is large and the Zeeman energy being small. Obviously, the QH skyrmion cannot be available in InSb 2DEGs with a relatively large g factor of -39 ~ -88 [New J. Phys. 13, 083010(2011)] that gives $k \gg 2$ at $B > 10$ T, which has been claimed in our previous paper [New J. Phys. 21, 083004(2019)].

Of course, it is also impossible to form the QH skyrmion at even filling $\nu = 2$ where the excitation gap is determined by the cyclotron energy. In this study, the simplest Ising (easy-axis) QHF (IQHF) occurs at the $\nu = 2$ LL crossing, which is more appropriate for investigating the domain wall (DW) dynamics of the QHF. There is a possibility for the realization of skyrmions once low-lying excited states of the DW as a 1D object have a nontrivial topological charge. This is the so-called DW skyrmion as being different from the QH skyrmion, a novel topological particle predicted in non-relativistic systems [e.g., in QHFs, PRL 82, 402(1999); in chiral magnets, PRB 99, 184412(2019)] and in relativistic systems [e.g., Nucl. Phys. B 872, 62(2013); J. Phys. A: Math. Theor. 46, 465401(2013)]. More recently, the experimental observation of DW skyrmions in chiral magnets with strong spin-orbit coupling (SOC) has been reported in preprints at <https://arXiv.org/abs/2004.07888> (2020) and <https://arxiv.org/abs/2004.06976> (2020). As a competing study, we have developed a novel RDNMR technique for the discovery of DW

skrymions in the $\nu = 2$ IQHF in this work. The related information was added in lines 27-32 and 48-50 in the revised version.

COMMENT 2: *Experimentally, nuclear-electron interaction has been studied in GaAs at $2/3$ for >20 years and is conventionally assigned to hyperfine interaction which is allowed due to the crossing of spin-up and down energy levels with no need to introduce a spin texture into domain walls.*

RESPONSE: It is generally believed that the dynamic nuclear polarization (DNP) induced by a current flow at $\nu = 2/3$ [corresponding to a composite-fermion (CF) filling factor $\nu^* = 2$] of GaAs 2DEGs is associated with domain structures at the LL crossing of CFs, but the underlying mechanism is still unclear [see a review paper in Semicond. Sci. Technol. 24, 023001(2009)]. The current-induced DNP is also assigned to polarize the nuclei in our $\nu = 2$ IQHF and two-subband IQHF in GaAs 2DEGs [PRL 98, 246802 (2007)]. Although the temperature-independent T_1 is found to be a common feature of these IQHFs characterized by *conventional* RDNMR measurements, there are also other important differences in the T_1 properties [New J. Phys. 21, 083004(2019)]: the T_1 measurement of the $\nu = 2/3$ IQHF gives a current-dependent T_1 between 180 s and 1350 s, while that of our $\nu = 2$ IQHF is relatively short (~ 45 s) and current-independent. T_1 of the two-subband IQHF is filling dependent in contrast to the ν -independent T_1 of our $\nu = 2$ IQHF. Furthermore, the thermally-activated transport measurements of both $\nu = 2/3$ and two-subband IQHFs suggests that low-lying energy excitations akin to skyrmions may dominate the nuclear spin relaxation (NSR) and thus T_1 . However, intricate CF-CF interactions at fractional fillings and an additional degree of freedom associated with the subband index complicate the interpretation.

In this study, we have developed a novel NMR technique called “NRDNMR” to characterize the *intrinsic* properties of the DW in the simplest IQHF at $\nu = 2$ that is not available by conventional methods. The T_1 measurement shows a very short $T_1 \sim 8$ s at $\nu = 2$ that increases with increasing temperature in sharp contrast to the results obtained from the *conventional* RDNMR measurement. Extensive and detailed analysis of the dependence of T_1 and T_2 on temperature, filling factor, quasiparticle localization, and effective magnetic fields allows us to interpret the ground state of the DW and its possible phase transitions in terms of Wigner solids of the DW skyrmion. We note that a continuous phase transition at zero effective magnetic fields shown in Fig. 3 indicates a direct link between the T_1 properties of the $\nu = 2$ IQHF obtained from the *conventional* RDNMR and NRDNMR measurements, which is well understood in the context of this interpretation. The related information was added in lines 128-130,148-153,163-169 in the revised version.

COMMENT 3: *This statement is rather confusing: either you have a gap between spin-up and down states and $R_{xx}=0$ or you have high enough T and/or disorder to mix the states and than $R_{xx}>0$. Either one or the other! Experimentally you see $R_{xx}>0$ which means the two energy levels are near-degenerate (at least the gap is $<$ disorder broadening) and there is no problem with energy conservation. These materials have much higher disorder than GaAs based on the width of the transition.*

RESPONSE: You are right that the merged peak is suggestive of energetically-degenerate two LLs, where the energy conservation required for the flip-flop process is satisfied. Indeed, the disorder broadening of LLs will even make the LLs with different spin indices overlap at integer ν , resulting in magnetoquantum oscillations of the NSR in a wide range of ν demonstrated in GaAs 2DEGs [PRL 64, 2563(1990)]. Similar results were obtained by our pump-probe NMR study of the $\nu = 2$ IQHF in InSb 2DEGs [New J. Phys. 21, 083004(2019)], where the NSR is shown to be dominated by the Korringa relaxation as being different from that of the $\nu = 2$ IQHF. However, the NSR at the merged peak was not observed because ν of this peak is beyond the range of probe filling factors controlled by gate voltages. Note that it is also difficult to measure the nuclear spin polarization (NSP) at this peak. For GaAs 2DEGs, an exponential relationship between R_{xx} and the activation or Zeeman energy gap Δ offers the possibility to perform the RDNMR measurement using the ESR or thermal polarization techniques because Δ and thus R_{xx} is sensitive to the Overhauser field B_N . For InSb 2DEGs, however, this relationship does not hold because the variable range hopping dominates the transport properties at both integer ν and the LL crossing (data not shown).

COMMENT 4: *Is there any EXPERIMENTAL evidence of the QHF formation rather than level crossing (apart from the fact that crossing point does not match single-particle crossing prediction, which it is not expected to)? In GaAs QHF state was identified by analyzing Knight shift in NMR.*

RESPONSE: The following results provide experimental evidence for the formation of the $\nu = 2$ IQHF in our InSb samples.

(1) The resistance spike near $\nu = 2$ as an additional peak in between two separate LL peaks (one locates at ~ 11 T in Fig.1c and the other above the maximum field of 15T in our measurements) is the signal of IQHF. In contrast, for a simple level crossing, the two approaching LLs will merge into one peak. Such an evolution of LLs as a function of magnetic fields (or filling factors) and tilt angles is clearly shown in our previous study [Phys. Rev. B 82, 241304(R) (2010)].

(2) The resistance spike near $\nu = 2$ is expected to originate from dissipative transport through a DW channel in the IQHF, where a first-order phase transition between the $P = 1$ and $P = 0$ domains occurs with zero effective magnetic field b^* [see, e.g., PRL 87, 216801(2001)]. Based on many-body Hartree-Fock theory, we have made a calculation of $b^* = 0$ in the B - B_{perp} plane (Supplementary Fig. 11) at which the IQHF is formed. It is shown that there is indeed an association between the theoretical $b^* = 0$ line and data points where the 1D DW channel percolating through the sample is expected to occur.

(3) As you mentioned, in GaAs QHF state was identified by analyzing Knight shift in NMR. The Knight shift of the RDNMR response near the spin-polarized ($P = 1$) domain relative to that near the spin-unpolarized ($P = 0$) domain in the $\nu = 2$ IQHF of our samples is about 35 kHz as measured in our previous study [Nat. Commun.8, 15084 (2017)].

In addition, the $\nu = 2$ IQHF with resistance spikes has also been formed in other 2DEGs with either large g factor [HgTe/CdHgTe, PRB 85, 245321(2012)] or large effective mass m^* [AlAs/AlGaAs, PRL 94, 176402(2005)] because the 2DEG sample with large values of m^*g is suitable for the tilted-magnetic-field experiment. We have reported the first demonstration of RDNMR in the simplest IQHF at $\nu = 2$ in InSb 2DEGs [PRB 82, 241304(R) (2010)]. We believe that the application of RDNMR measurements should be extended to the above-mentioned 2DEGs.

The related information was added in lines 73-78.

COMMENT 5: *Is there is any evidence of non-collinear spin arrangement in these domain walls (or any other in QHE experiments)? I do not know of either theoretical nor experimental claims. With such a large g -factor in InSb spin textures will be very energetically unfavorable.*

RESPONSE: Theoretical study of the DW skyrmion in the $\nu = 2$ IQHF has been reported in PRB 66, 041308(R) (2002) and PRB 100, 235406 (2019). Although the $\nu = 2$ IQHF was formed in 2DEGs with either large g factor (InSb/AlInSb and HgTe/CdHgTe) or large effective mass m^* (AlAs/AlGaAs) as mentioned above, the RDNMR measurement was applied only to the $\nu = 2$ IQHF of the InSb 2DEG so far [PRB 82, 241304(R) (2010)]. In this study, we have developed a novel RDNMR technique combined with the NSR measurement as a generally accepted methodology for determining the low-frequency spin textures in the QH system (e.g., QH skyrmions) to demonstrate the DW skyrmion in the $\nu = 2$ IQHF of the InSb 2DEG.

As discussed in Response to Comment 1, the large g factor due to a strong SOC in the InSb 2DEG results in a large bare Zeeman splitting that prevents the formation of QH skyrmion. However, the SOC in the IQHFs is off-diagonal with respect to the LL basis and treated as a perturbation [PRL 82, 402(1999)]. It twists the planar component of the polarization vector and lifts the degeneracy of the DW ground state, allowing a topological excitation of the DW skyrmion to be the relevant charged particle whose size and energy are dependent of the SOC strength. Furthermore, the electron transport between domains with opposite spin polarization is known to be accompanied by the spin-flip process that is usually mediated by the hyperfine interaction or SOC in order to conserve the angular momentum. Although compared to the hyperfine interaction the SOC is much more efficient to flip electron spins, the role played by the nuclear spins is still evident from the observed RDNMR signal in our study. This further approves the perturbative treatment of SOC in the IQHF.

In experiments the NSR in the $\nu = 2$ IQHF is distinctly different from the Korringa relaxation mechanism for a simple level crossing, allowing us to explain the data by referring to low-lying spin textures. Extensive and detailed analysis of the dependence of T_1 and T_2 on temperature, filling factor, quasiparticle localization and effective magnetic fields enables us to interpret the ground state of the DW and its possible phase transitions in terms of Wigner solids of the DW skyrmion. More recently, the DW skyrmion has been demonstrated [<https://arXiv.org/abs/2004.07888> (2020) & <https://arxiv.org/abs/2004.06976> (2020)] in chiral magnets with a strong DMI induced by SOC. The related information was added in lines 39-40, 190-194.

COMMENT 6: *Hyperfine interaction does not require tilted electron spins since at this T and B equilibrium nuclear polarization is just a few percent and nuclear spins point in all directions.*

RESPONSE: The $\nu = 2$ IQHF in InSb 2DEGs is a highly sensitive region for the RDNMR measurement as analyzed in our previous study [Nat. Commun. 8, 15084(2017)]: a shift of the IQHF spike caused by the DNP with the degree of nuclear polarization of $\sim 10\%$ (This information was added in lines 131-133) allows detection of the RDNMR signal with a ratio of resistance changes up to a few percent, as similar to the RDNMR measurement of the $\nu = 2/3$ IQHF in the

GaAs 2DEG. Therefore, the RDNMR measurement provides a local probe of nuclei dynamically polarized near the DW opposite to their equilibrium direction. Although the current-induced DNP in the DW does not require tilted electron spins, the NSR therein as being different from the Korringa relaxation mechanism for a simple level crossing (see responses to Comments 2 & 3) strongly suggests that low-lying excited spin textures should be responsible for the relaxation process.

COMMENT 7: *Lowest spin excitations are not magnons but magneto-rotons with $E < E_z$, see e.g. 10.1103/PhysRevLett.107.066804.*

RESPONSE: It is known that magnetorotons as a dispersion of collective excitations possessing a minimum in energy at large momenta (or short wavelength) govern the low-temperature thermodynamics of the electron system in the *fractional* quantum Hall regime. However, our work focuses on the electron system in the *integer* quantum Hall regime and on the effect of long-wavelength spin excitations on the NSR investigated by NMR. The simplest low-lying excitations out of the ground state in a uniform QHF are spin waves (magnons) that have a minimum excitation gap given by the Zeeman splitting. Because the nuclei precess at frequencies about three orders of magnitude below that of the Zeeman splitting, they do not couple effectively to ordinary spin waves. Therefore, we note that these spin waves cannot accelerate the NSR.

In conclusion, the IQHF with domain structures rather than a simple level crossing occurs at $\nu = 2$ in our samples as supported by both transport and NMR results. The diversity in the T_1 properties of the $\nu = 2$ IQHF obtained from both RDNMR and NRDNMR measurements cannot be explained by the scenario of simple level crossing dominated by the Korringa relaxation mechanism despite its implications for the hyperfine interaction. *All* findings and conclusions presented in this manuscript have been well justified in terms of Wigner solids of the DW skyrmion. We hope our replies will clear up any doubts you may have.

Anyway, thank you to let us know your concerns. More detailed information addressing these concerns in the revised version will help improve the manuscript. Thank you for taking the time and effort to complete the review.

2. RESPONSES TO REFEREE #2's COMMENTS:

We're pleased to hear that you find this work important and recommend publication in Nature Communications. We gratefully appreciate your constructive comments and suggestions that help improve the manuscript.

Reviewers' Comments:

Reviewer #1:

Remarks to the Author:

In response to my questions authors provided long and in most cases satisfactory answers. There are a few important points, though, which I do not agree with.

In answering my question 1, authors agree that regular skyrmions cannot be formed in InSb due to the large g-factor but refer to the recent studies in Co-based chiral magnetic materials where DW skyrmions have been observed. As far as I know these materials have small g-factor close to 2 (in Co it is ~ 2.5), so the question whether DW skyrmions are energetically permitted in $g > 40$ material is still a valid concern.

In a previous work [New J. Phys. 21, 083004(2019)] authors studied spin relaxation in a 2D gas in a similar 2D gas from a similar heterostructure using more conventional RDNMR technique where $T_1 \sim 45$ s and almost no T-dependence has been measured (already in that paper authors speculated that DW skyrmions may be involved). In the current work authors use a "novel" NRDNMR method and measured lower T_1 and decrease of $1/T_1$ with increase of T which are also being associated with DW skyrmions. There is no discussion of this difference in the manuscript despite samples seem to be very similar.

The reason for the large signal in NRDNMR configuration is exponential (or strongly non-linear) dependence of non-local resistance on resistivity and bulk-edge coupling. In the paper authors treat NRDNMR signal as if it would be simply proportional to σ_{xx} with some constant coefficient. Non-local resistance R_{NL} is a function of (unknown) coupling between edge states $t(T, I)$ and the bulk conductivity of the topmost LL σ_{xx} , R_{NL} temperature and current dependence includes T- and I-dependence of this coupling (see e.g. V. Goldman, PRB from 1992). Thus, the use of $R_{NL}(T) \propto \sigma_{xx}(T)$ and subsequent fit of $R_{NL}(T)$ with variable range hopping is not proper. The same $R_{NL}(T) \propto \sigma_{xx}(T)$ is implied in the analysis of Fig. 2c and the claim "An increase in $1/T_1$ with decreasing $R_{14,23}$ suggests that quasiparticle localization rather than additional conducting states dominates the NSR" is poorly justified.

To summarize the data, in the relevant regime (Phase I) the data measured using "novel" NRDNMR is not following a Koringa law for 3D systems $1/T_1 \propto T$. The values of T_1 are much smaller than the values measured in GaAs and in the same materials InSb using conventional RDNMR (where resistive response is proportional to the bulk conductivity), no explanation for the discrepancy for the same material is given. The authors argue that the observed reduction of T_1 requires low energy spin excitations and invoke a crystal of DW skyrmions formation. At the moment I still not convinced that T_1 and its T-dependence extracted from a non-local resistance response is representative of the bulk (and DWs) T_1 response, and whether non-Koringa T_1 is a signature of a skyrmion crystal in 1D systems.

RESPONSES TO REVIEWER #1's COMMENTS:

We're pleased to hear that you find our last responses to your questions provide satisfactory answers in most cases. We want to extend our thanks and appreciation for your time and effort in reading and commenting the manuscript. Below we provide a point-by-point response to the remaining concerns.

COMMENT 1: *In answering my question 1, authors agree that regular skyrmions cannot be formed in InSb due to the large g-factor but refer to the recent studies in Co-based chiral magnetic materials where DW skyrmions have been observed. As far as I know these materials have small g-factor close to 2 (in Co it is ~2.5), so the question whether DW skyrmions are energetically permitted in $g > 40$ material is still a valid concern.*

RESPONSE: In contrast to the case of QH skyrmions whose energy is determined by the effective g factor g^* in the form of the Zeeman energy E_z , the energy of DW skyrmions in the IQHF is determined by the SOC strength in a perturbative form of $\alpha = \epsilon_{so}/\hbar\omega_c$ [see Ref. 6 and parameters given in Table 1]. Note that g^* depends not only on the SOC strength but also on other parameters (e.g., band gap, band matrix element, exchange interaction, etc.). It is shown in Table 1 that α in InSb 2DEGs is only 3.8 times larger than that in GaAs 2DEGs while the difference in E_z between these two 2DEGs is significant (over 100 times). The related information was added in lines 194-200 and in Supplementary Table 1.

Similarly, the SOC rather than the g factor contributes to a chiral DMI [Nat. Nanotech. 8, 152 (2013) and Nat. Rev. Mater. 2, 17031 (2017)] that stabilizes the DM skyrmion in magnetic systems.

Table 1. Comparison of Zeeman energy and perturbative ratio α in GaAs 2DEGs and InSb 2DEGs (well width $w = 20$ nm, $B = 13$ T, magnetic length $l_B = 256\text{\AA}/\sqrt{B}$). Parameters are taken from R.Winkler: *Spin-orbit coupling effects in two-dimensional electron and hole systems* (Springer, Berlin, 2003) and from our experimental data.

	GaAs	InSb
effective g factor	-0.44	-54
Zeeman energy E_z (meV)	0.3	40
effective mass m^* (in units of m_0)	0.067	0.015
cyclotron energy (meV) $\hbar\omega_c = \hbar \frac{eB}{m^*}$	22.4	100
SOC coefficient γ (eV \cdot \AA^3)	27	500
SOC energy (meV) $\epsilon_{so} = \gamma \left(\frac{\pi}{w}\right)^2 / l_B$	0.1	1.7
perturbative ratio α	0.0045	0.017

COMMENT 2: *In a previous work [New J. Phys. 21, 083004(2019)] authors studies spin relaxation in a 2D gas in a similar 2D gas from a similar heterostructure using more conventional RDNMR technique where $T_1 \sim 45$ s and almost no T -dependence has been measured (already in that paper authors speculated that DW skyrmions may be involved). In the current work authors use a "novel" NRDNMR method and measured lower T_1 and decrease of $1/T_1$ with increase of T which are also being associated with DW skyrmions. There is no discussion of this difference in the manuscript despite samples seems to be very similar.*

RESPONSE: In the last response letter, we have pointed out that a continuous phase transition at zero effective magnetic fields b^* shown in Fig. 3 indicates a direct link between the T_1 properties of the $\nu = 2$ IQHF obtained from the NRDNMR measurement (T -independent $T_1 \sim 40$ s at $|\nu - 2| > 0.05$, Fig. 3b) and those obtained from the RDNMR measurement [T -independent $T_1 \sim 45$ s, New J. Phys. 21, 083004(2019)]. This result strongly suggests that the presence of *1D Wigner glass* of the DW skyrmion is responsible for the nuclear spin relaxation in the $\nu = 2$ IQHF characterized by the RDNMR measurement (lines 247-274). Note that the four-terminal resistance $R_{kl,mn}$ in both NRDNMR and RDNMR measurements of the $\nu = 2$ IQHF is only parameterized by a longitudinal resistance of the bulk state (Supplementary Notes 1&2 and see also below), and in this sense all these resistances are essentially identical. Therefore, the T_1 response obtained from the time dependence of DNP-induced changes in $R_{kl,mn}$ is indicative of the bulk (DW) state for both measurements.

By the way, the key aspects of our newly-developed NRDNMR technique lie in relatively low nonlocal resistances that respond more sensitively to subtle changes in the sample (lines 60-70). This makes it possible to measure the effect of DNP at very low current corresponding to a small Hall electric field in the bulk, thus allowing for the study of the DW structure under *equilibrium* conditions that accounts for the difference in the T_1 properties characterized by the NRDNMR and RDNMR measurements.

COMMENT 3: *The reason for the large signal in NRDNMR configuration is exponential (or strongly non-linear) dependence of non-local resistance on resistivity and bulk-edge coupling. In the paper authors treat NRDNMR signal as if it would be simply proportional to σ_{xx} with some constant coefficient. Non-local resistance R_{NL} is a function of (unknown) coupling between edge states $t(T,I)$ and the bulk conductivity of the topmost LL σ_{xx} , R_{NL} temperature and current dependence includes T - and I -dependence of this coupling (see e.g. V. Goldman, PRB from 1992). Thus, the use of $R_{NL}(T) \propto \sigma_{xx}(T)$ and subsequent fit of $R_{NL}(T)$ with variable range hopping is not proper. The same $R_{NL}(T) \propto \sigma_{xx}(T)$ is implied in the analysis of Fig. 2c and the claim "An increase in $1/T_1$ with decreasing $R_{14,23}$ suggests that quasiparticle localization rather than additional conducting states dominates the NSR" is poorly justified.*

RESPONSE: It is right that both the bulk conductivity of the topmost LL and the edge-bulk scattering may contribute to the nonlocal resistance. A decrease in nonlocal resistance with increasing temperatures due to the phonon-assisted inter-edge-bulk scattering (Goldman, 1992) shown in the temperature dependence of R_{LL} at the LL peak (Fig. 1d) suggests a finite amount of scattering between the topmost (bulk) LL and the edge channel corresponding to its nearest neighboring LL. For the $\nu = 2$ IQHF with LL intersections, however, the bulk state has domain

structures where the edge states corresponding to the two intersecting LLs (the nearest neighboring LLs of the bulk state) become part of an array of (bulk) domains [our work in Nat. Commun.8, 15084 (2017)]. In this case, the current is mainly carried by the edge channel related to the lowest LL (the *next*-nearest LL of the bulk state) with negligible scattering to the bulk state. This conclusion is supported by the data in Fig. 1d where the thermal response of FSs in the IQHF is opposite to that of R_{LL} . The data is fit well by the VRH formula, strongly suggesting that the FSs are dominated by the bulk properties of the IQHF. Furthermore, the current dependence of these FSs (Fig. 1f) is also different from that of R_{LL} (Goldman, 1992). In this work, the four-terminal resistance $R_{kl,mn}$ of the $\nu = 2$ IQHF is only parameterized by a longitudinal resistance of the bulk state using the bulk-edge model with independent edge- and bulk-components (Ref. 21 and Supplementary Notes 1&2). Therefore, $R_{NL} \propto \sigma_{xx}$ and related analyses are justified. The related information has been added in lines 91-117.

COMMENT 4: *To summarize the data, in the relevant regime (Phase I) the data measured using "novel" NRDNMR is not following a Korringa law for 3D systems $1/T_1 \propto T$. The values of T_1 are much smaller than the values measured in GaAs and in the same materials InSb using conventional RDNMR (where resistive response is proportional to the bulk conductivity), no explanation for the discrepancy for the same material is given. The authors argue that the observed reduction of T_1 requires low energy spin excitations and invoke a crystal of DW skyrmions formation. At the moment I still not convinced that T_1 and its T -dependence extracted from a non-local resistance response is representative of the bulk (and DWs) T_1 response, and whether non-Korringa T_1 is a signature of a skyrmion crystal in 1D systems.*

RESPONSE: The difference in T_1 characterized by the NRDNMR and RDNMR measurements and the T_1 response to the bulk properties of the IQHF have been addressed above. Extensive and detailed analysis of the dependence of *both* T_1 and T_2 *not only* on temperature *but also* on filling factor, quasiparticle localization and effective magnetic fields enables us to interpret the ground state of the DW and its possible phase transitions in terms of Wigner solids (crystal and glass) of the DW skyrmion. The unique temperature dependence of T_1 observed in both 1D DW skyrmion crystals (this work) and 2D skyrmion crystals (Ref. 33) requires future theoretical work.

Reviewers' Comments:

Reviewer #3:

Remarks to the Author:

The results, considered at the level of data are certainly noteworthy and are a significant addition to the field. I have looked at the questions posed by other reviewers and find the authors responded satisfactorily. There is certainly a phase of interest near the QHFM point, and the authors' arguments that this phase is a Wigner solid of some type are reasonable. The spectacular phenomena and that justifies publication in this journal to my mind.

Much of the detail of the paper is hard to discern from the presentation. For example it seems (mostly from the comments to other reviewers) that the range of low T1 is apparent in only a limited number of measurement configuration (maybe just one). While an important point related to the repeatability of the experiment, and one that should be made clearer in the paper, I do not believe that the point detracts from the conclusions. It is clear that the phenomenon of interest is delicate, requiring as it does, nonlocal measurements.

The paper suffers in my opinion from the constraint of space due to the journal format. I would recommend more judicious use of the supplementary information, as is often done for this journal, or a change to a longer-format article. For example b^* , is of crucial importance and is presented essentially without definition other than to call it "effective field". More of a definition of the variable should have been provided; it would be standard only for people with deep familiarity with QHFM work. Effective field often means something else in DNP experiments.

My suggestion: much of the description of nonlocal measurement could have been put into summary, leaving more room for presentation and discussion of NMR data and discussion of phases. I would like to see some discussion of length scales associated with the Wigner solids, even skyrmion separation, and if possible characteristic size of the network of DW's. The DWS will Coulomb interact: can you really say the Wigner solid is 1 D if the DWs are apart by a distance similar to that between skyrmions along a single DW.

RESPONSES TO REVIEWER #3's COMMENTS:

We're pleased to hear that you find this work important and recommend it to be published in Nature Communications. We gratefully appreciate your constructive comments and suggestions that help improve the manuscript. Below we provide a point-by-point response to these concerns.

COMMENT 1: *It seems (mostly from the comments to other reviewers) that the range of low T1 is apparent in only a limited number of measurement configuration (maybe just one). While an important point related to the repeatability of the experiment, and one that should be made clearer in the paper, I do not believe that the point detracts from the conclusions. It is clear that the phenomenon of interest is delicate, requiring as it does, nonlocal measurements.*

RESPONSE: The statement regarding the repeatability of our experiment was added in lines 318-323 in the revised version with the track changes feature.

COMMENT 2: *I would recommend more judicious use of the supplementary information, as is often done for this journal, or a change to a longer-format article. For example b^* , is of crucial importance and is presented essentially without definition other than to call it "effective field". More of a definition of the variable should have been provided; it would be standard only for people with deep familiarity with QHFM work. Effective field often means something else in DNP experiments.*

RESPONSE: We have provided the Supplementary Note 3 for the definition and calculation of effective fields.

COMMENT 3: *My suggestion: much of the description of nonlocal measurement could have been put into supplementary, leaving more room for presentation and discussion of NMR data and discussion of phases. I would like to see some discussion of length scales associated with the Wigner solids, even skyrmion separation, and if possible characteristic size of the network of DW's. The DWS will Coulomb interact: can you really say the Wigner solid is 1 D if the DWs are apart by a distance similar to that between skyrmions along a single DW.*

RESPONSE: The description of nonlocal measurements has been put into Supplementary Notes 1 and 2. Discussion of length scales of DW's and dimensionality of the Wigner solid of DW skyrmions was added in lines 305-312 in the revised version with the track changes feature.